# The Role of L-Glutamate as an Umami Substance for the Reduction of Salt Consumption: Lessons from Clinical Trials

**DOI:** 10.3390/nu17101684

**Published:** 2025-05-15

**Authors:** Hideki Matsumoto, Licht Miyamoto, Takaki Matsumoto, Francois Blachier

**Affiliations:** 1Department of Nutrition and Life Science, Faculty of Health and Medical Sciences, Kanagawa Institute of Technology, Atsugi 243-0292, Kanagawa, Japan; 2Faculty of Science, Ehime University, Matsuyama 790-8577, Ehime, Japan; 3UMR Nutrition Physiology and Alimentary Behavior, Université Paris-Saclay, AgroParisTech, INRAE, 91120 Palaiseau, France; francoismichel.blachier@gmail.com

**Keywords:** salt reduction, appropriate salt intake, sodium restriction, glutamate, quality of life

## Abstract

Salt as sodium chloride is an essential mineral present in food which is involved in physiological functions such as nutrient intestinal absorption, nerve conduction, and muscle contraction. It plays a critical role in food flavoring and ingestive behavior, serving as the basis of one of the five basic tastes. However, excessive salt intake is widely recognized as a risk factor for lifestyle-related diseases, such as hypertension, making salt reduction a key strategy in terms of public health. In that overall context, the aim of this review is to recapitulate the various approaches for salt intake reduction which have been implemented, with a focus on the use of L-glutamate in umami as a sodium substitute. Umami substances, like salt, are one of the five basic tastes and have the potential to enhance the flavor of food while simultaneously reducing salt intake. Several clinical trials have shown that L-glutamate can compensate for the reduction in saltiness while improving the overall palatability of food. This characteristic makes umami substances a valuable element in the context of salt reduction. By incorporating L-glutamate into the diet, it becomes possible to maintain a balanced nutritional intake while reducing salt, making it an effective approach toward a healthier diet. At the same time, L-glutamate-induced salt intake reduction potentially alleviates stress-related indicators associated with salt reduction. Thus, the strategic use of L-glutamate as compound involved in umami taste can help compensate for changes in taste perception due to salt reduction, enabling individuals to maintain meal satisfaction while transitioning to healthier dietary habits with lower salt.

## 1. Introduction

Salt as sodium chloride (NaCl) is well known to play a central role in flavoring food and enhancing its taste and palatability, as well as for food preservation [1]. Salty represents one among the five basic tastes, which are sweet, salty, sour, bitter, and umami.

Salt is present in numerous processed foods including meat and cheese, as well as in condiments and sauces and in table salt. The sodium ion present in NaCl is involved in numerous vital physiological functions, including the intestinal absorption of amino acids and glucose [2], the transmission of nerve impulses [3], and muscle contraction [4]. Generally, when following a normal diet, sodium deficiency is rarely observed. However, excessive sweating, repeated episodes of severe diarrhea, or vomiting can cause sodium to be lost from the body, leading to a state of salt deficiency (hyponatremia). In such cases, symptoms such as fatigue, loss of appetite, headache, nausea, dizziness, lightheadedness, and muscle cramps may occur [5,6,7].

On the other hand, excessive salt intake is recognized as a risk factor for high blood pressure and, consequently, for cardiovascular diseases [8]. In fact, an increased concentration of sodium (hypernatremia) in the body leads to increased water retention, which is associated with symptoms such as elevated blood pressure, dizziness, and headaches [9,10]. In addition, symptoms and diseases related to the cardiovascular system (hypertension, cardiac hypertrophy, vascular damage, renal damage, heart failure, ischemic heart disease, stroke, renal failure) as well as those related to the non-cardiovascular system (urolithiasis, osteoporosis, gastric cancer, bronchial asthma) have been reported [11,12,13,14,15,16,17,18,19]. From these data, an appropriate salt (sodium) intake is of prime importance, and guidance on salt intake during hyponatremia and salt reduction during excessive intake are considered crucial measures and policies as contributors of health maintenance (Figure 1).

Efforts to achieve appropriate salt intake have been carried out in various countries around the world. Salt supplementation measures are implemented during hyponatremia, but on the contrary, salt reduction measurements and policies aimed at preventing excessive salt intake are carried out from the perspective of preventing lifestyle-related diseases. The reduction in salt intake is strongly recommended by the World Health Organization (WHO) to lower the risk of some non-communicable diseases (NCDs) [20,21,22]. Therefore, the WHO has set a target to reduce salt intake by 30% between 2011 and 2025 as part of the NCD Global Monitoring Framework [23]. To achieve these objectives, the joint WHO/food and agriculture organization (FAO) expert consultation recommends a target average salt intake of 5 g per day for adults [20,21,22], while the American College of Cardiology (ACC) and American Heart Association (AHA) guideline in 2017 recommend salt intake to be less than 3.8 g/day [24,25]. Regarding the European Society of Hypertension–European Society of Cardiology (ESH-ESC), their 2018 guideline recommends a salt intake not above 5 g/day [26,27]. In Japan, the target value recommended by the Japanese Society of Hypertension (JSH) in their 2019 guidelines for hypertension treatment is set to less than 6 g/day [28]. While the recommended salt intake differs across various countries due to different national policies, in practice, the actual salt intake in most countries exceeds the recommended amounts [29]. As a result, various efforts to reduce salt intake are promoted in each country.

Various methods have been explored and implemented to reduce salt intake, especially through efforts to decrease the amount of salt consumed from food [30]. While these efforts successfully lower salt content, they also result in a reduction in the salty taste in food, thus reducing food palatability and consequently affecting food intake. Reduced food intake may lead in various subpopulations to insufficient calorie intake, notably in individuals characterized by protein–energy malnutrition [31]. Furthermore, the compliance for reduced salt intake in the long term appears poor [32,33]. In addition, long-term attempts to reduce salt intake are effective, but it has been pointed out that overly strict salt reduction should be carried out with caution due to potential physical negative effects, mainly caused by insufficient calorie intake [34]. As an alternative to these limitations, efforts have been made to preserve the taste of reduced-salt dishes by incorporating umami (broth), acidity (citrus), spiciness (spices), and texture (roast). By providing reduced-salt meals that do not compromise food taste and palatability, it is likely that the compliance for such dietary restriction would be ameliorated.

L-glutamate (Glu), which was first discovered from a plant protein, namely wheat gluten, by Ritthausen, K. H. in 1866 [35], is a dispensable (non-essential) amino acid that can be synthesized by the body [36]. L-Glu is present in foods both as a protein-bound and free amino acid, with free L-Glu being particularly abundant in tomatoes, cheese, cured ham, vegetables, kelp, and human breast milk. The free form of L-Glu is present in various salt forms (sodium, calcium, potassium, ammonium, or magnesium, etc.) in foods. L-Glu as an umami substance was discovered by Ikeda, K in 1908. The first paper reporting this discovery was published in 1909 [37]. L-Glu has been used as a seasoning for over a century, with its sodium salt being the most commonly used form among the various salts. L-Glu has been assessed by JECFA multiple times for its safety of use as a flavor enhancer, and it has been concluded that it poses no safety concerns when used at normal levels [38]. Furthermore, the U.S. Food and Drug Administration (FDA) designated L-Glu as a Generally Recognized As Safe (GRAS) substance in 1958, along with many other common food ingredients such as salt, vinegar, and baking powder, under regulation No. 182.1(a). On another hand, L-Glu is converted into α-ketoglutarate by enzymes such as glutamate–pyruvate transaminase (GPT) and glutamate dehydrogenase (GLDH), supporting cellular energy metabolism through its entry into the tricarboxylic acid (TCA) cycle. Additionally, its functions in cell signaling as a ligand for glutamate receptors and as an excitatory neurotransmitter have been documented.

In such a context, this review summarizes the capacity of L-Glu, an umami substance, for the reduction in salt intake, as evidenced in various experimental and clinical situations, along with the underlying mechanisms. The present review also considers the effects of L-Glu in terms of sensory assessments, stress assessments, and food intake. The primary focus of this review is related to the assessment of the efficacy of L-Glu as a food ingredient for salt reduction in relationship with indices of health benefits, quality of life (QOL), and well-being.

## 2. The Position of Saltiness and Umami Among the Elements of Pleasant Taste

The five senses, namely taste, smell, touch, eyesight, and hearing, along with individual experiences, physical condition, atmospheric conditions, and external factors such as food culture, are involved in the brain assessment of palatability and in ingestive behavior [39]. Throughout human history, seasonings and the use of ingredients, as well as cooking techniques, have been used to improve the eating environment (Figure 2) [40].

Among the different parameters involved in palatability, the basic tastes (thus sweetness, sourness, saltiness, bitterness, and umami) are the most important. Both saltiness and umami are known to be important elements that contribute to the thus named “deliciousness of food” [41]. As substances that exhibit saltiness, sodium chloride (NaCl) found in seawater and rock salt has been used for thousands of years [42,43]. Substances that exhibit umami taste include L-Glu as an amino acid, and its salts in the form of sodium, potassium, calcium, magnesium, ammonium salts, as well as 5′-ribonucleotides which belong to the nucleic acid compounds (Figure 3).

L-Glu plays important roles as a building block of proteins, as an energy substrate, and in metabolism, as well as being an umami substance [44,45]. It exists in two forms: as a building block of proteins and in its free form in the body. The L-Glu in its building block form is an abundant amino acid in dietary proteins [46]. The free form of L-Glu functions as an umami substance and exists in various salt forms in food [47]. In fact, four types of glutamate salts, monosodium glutamate (MSG), mono potassium glutamate (MPG), calcium di-glutamate (CDG) and magnesium di-glutamate (MDG), have been reported to exhibit the same level of umami substance as determined in sensory evaluations [48]. The free-form L-Glu is found in foods we commonly consume (such as cheese, tomatoes, and broccoli) and affects the taste of these foods [49,50,51]. Furthermore, cooking methods that concentrate the free-form L-Glu are used in fermented foods such as cheese, in dashi, which is essential in Japanese cuisine, as well as in soup stocks for Western and Chinese cuisines [52]. Incidentally, the free form L-Glu is utilized mainly in its monosodium salt form (MSG), which is used as a food exhauster for human nutrition as well as a flavor enhancer as an umami substance [53]. Additionally, breast milk contains more than half of the free-form amino acids as L-Glu, and thus mammalian newborns, including humans, consume L-Glu from breast milk shortly after birth [54,55,56]. In this way, the ability to perceive saltiness and umami as tastes plays an early and important role in human ingestive behavior [57].

## 3. Physiological Roles of Saltiness and Umami for Nutrition Intake

Briefly, taste plays the role of a sensory system that contributes to alerting the organism of the presence of essential nutrients necessary for survival and harmful substances to avoid. Among these, the taste of salt is believed to indicate in the diet essential minerals required by the organism, while umami is recognized as signifying amino acids and proteins in food, with these latter compounds providing energy and nitrogen sources [58,59].

In fact, as shown in Figure 4, the salt component Na in food transmits information to the brain through its interaction with salt taste receptors located on tongue surface (taste bud) in the mouth [60], where the sensation of saltiness is recognized. Additionally, Na absorbed into the body plays an important role in maintaining the osmotic pressure and mineral balance within the body. On the other hand, the umami substance L-Glu transmits information to the brain through its interaction with L-Glu receptors in the oral cavity (in the taste bud) and in the mucosa of the digestive tract, where the sensation of umami is then recognized [61]. In turn, the brain sends signals to peripheral organs to prepare for the efficient digestion of food. In fact, L-Glu triggers physiological responses such as the promotion of saliva secretion in the mouth [62] and digestive fluid secretion in the stomach [63], which contribute to protein digestion [64]. Thus, both the salt substance Na and the umami substance L-Glu are not only taste substances but also essential nutrients that play important roles in the body’s physiological functions.

## 4. Various Cooking Methods to Support the Implementation of Reduced Salt Intake

In efforts to reduce salt intake, the decrease in saltiness, one of the factors that contribute to the overall pleasantness of a meal (Figure 2), ultimately affects the overall taste. Thus, simply reducing salt intake can impact the overall taste of the meal. To reduce salt intake without diminishing the deliciousness of food, efforts have been made throughout the history of humanity to adjust various factors that contribute to taste, aside from saltiness (Figure 2). The main strategies include (1) physically reducing the amount of salt consumed by reducing food volume; (2) using ingredients and seasonings with distinctive flavors when using less salt; and (3) innovating cooking methods to reduce salt intake. In the first approach, methods include the reduction in calorie intake while keeping the salt content unchanged (preservation of taste) or reducing the salt content (making the taste milder) without changing portion sizes. In the second approach, utilizing sourness, spices, and flavorings (such as lemon, yuzu, chili pepper, sansho, wasabi, sesame oil, etc.), as well as umami (contained in food such as powder cheese, tomato paste, dashi, mushrooms, and umami seasonings), are examples of such techniques. The third approach involves the utilization of changes in the visual appearance, texture, and cooking techniques (such as browning, frying, or enhancing aroma) to create appealing dishes [65].

These various efforts to reduce salt intake represent the wisdom humanity has accumulated through experience, and they should be preserved and passed down for future generations, notably when they have proved to be beneficial for human health and well-being. However, many of these methods lack scientific trials and reports of their result evaluation.

## 5. Studies on the Application of Umami’s Physiological Effects for Salt Reduction

The efforts to reduce salt intake through the use of umami substances have been reported in strategic plans, reviews, and studies, such as those by the Expert Committee on Strategies for Reducing Sodium Intake in the United States [66], in the study by Prabhavathis et al. [67], Nomura et al. [68], and in the study by Tanaka et al. [69]. Currently, as will be presented in the following chapters, some studies have shown that it is possible to maintain food palatability with a lowered overall sodium level in a given food when salt is replaced by L-Glu.

### 5.1. A Sensory Evaluation of Samples with Different Na and Glu Contents in a Clear Soup Such as Chicken Soup or Other Aqueous Solutions

In this section, we present a compilation of studies that employed sensory evaluation to assess the impact of umami substances on perceived taste intensity in response to variations in salt concentration (Table 1). According to a report by Yamaguchi et al., by using sensory evaluation methods [70], the palatability of a 0.7% (*w*/*v*) sodium chloride (NaCl) solution decreases when the concentration is reduced to 0.4% (*w*/*v*) NaCl. However, by adding 0.38% (*w*/*v*) Glu as an umami substance to the 0.4% (*w*/*v*) NaCl solution, the palatability was found to be equivalent to that of the 0.7% (*w*/*v*) NaCl solution before salt reduction. This result demonstrates that by using Glu, it is possible to reduce salt intake by about 40% without compromising palatability. Similar results have also been reported by Roininen et al. (with 44 participants in this study) [71], as well as in the study by Ball et al. (with 120 participants) [72], in the study by Cater et al. (with 34 participants) [73], and finally in the study by Morita et al. (with 11 participants) [48], establishing robustly the reality and reproducibility of this phenomenon (Figure 5).

Furthermore, in a report by Hayabuchi et al. [74], a multicenter randomized double-blind trial was conducted with 584 participants across seven facilities to evaluate the effect of 0.3% (*w*/*v*) L-Glu on the palatability of 0.3%, 0.6%, and 0.9% (*w*/*v*) NaCl solutions. In this study, the sensory evaluation scores for palatability were significantly lower for the 0.3% (*w*/*v*) NaCl solution, which had approximately 50% less salt compared to the 0.6% (*w*/*v*) NaCl solution. However, when 0.3% (*w*/*v*) L-Glu was added to the 0.3% (*w*/*v*) NaCl solution, the palatability score was found to be equivalent to that of the 0.6% (*w*/*v*) NaCl solution before salt reduction (Figure 5). This result also reported that, under the conditions of the multicenter randomized double-blind trial conducted with healthy adults, thus under the condition of utilization of a clinically reliable method, the palatability of low-NaCl aqueous solutions was enhanced by the addition of the umami substance L-Glu.

### 5.2. Sensory Evaluation of a Menu with Different Na and Glu Contents in a Food-as-Set-Out and Sodium-Reduction Intervention Study

As summarized in Figure 5, the utilization of L-Glu as umami substance in a sodium-reduction diet in different studies was shown, through sensory evaluation methods, to not compromise palatability even with a marked reduction in the dietary salt content. On the other hand, there are few reports investigating the effects of overall salt reduction in daily meals on food palatability. Furthermore, we do not know if this reduction can be maintained in the long term. Accordingly, in this section, the first part presents a study on the development of sodium-reduced hospital meals designed to maintain palatability through the use of umami seasonings, as evaluated by sensory testing. The second part presents findings from an intervention study using these meals in a reduced-sodium dietary regimen for hospitalized patients (Table 1).

In a report by Ishida et al. [75], magnesium di-glutamate (MDG) was used as an umami substance to reduce the salt content of regular hospital meals, and accordingly a reduced-salt meal was prepared to achieve the same sensory evaluation scores as the regular meal before reduction. A report by Kawano et al. [72] from the same hospital describes a 14-day single-blind crossover intervention study conducted on hospitalized patients using the regular meal and a reduced-salt meal with added umami substances (14-32% less salt compared to the regular meals). The results show that overall, the dietary interventions allow us to reduce the daily salt intake on all days from the first to the fourteenth day of the intervention. The average daily salt intake over the 14-day intervention period was indeed 3.28 g/day for the regular meals and 2.43 g/day for the reduced-salt meal with umami substances (MDG added). The average daily salt-reduction rate for the entire diet was as much as 26%. Additionally, the energy intake (the primary evaluation item in the study)—a challenge in continuing salt-reduction therapy—remained the same before and after the intervention under the study conditions. Indeed, the average daily calorie intake over the 14 days was 1505 kcal for the regular meal and 1621 kcal for the MDG-added reduced-salt meal, showing no statistically significant difference in calorie intake. Furthermore, the intake of protein, carbohydrates, and fats was equivalent both before and after the intervention and between the groups.

In these reports, reduced-salt menus utilizing umami substances were prepared [75], and as shown in Figure 6, by combining them, a daily meal plan was structured, enabling continuous salt reduction [76] while maintaining calorie intake. Thus, by using umami substances in reduced-salt meals on the daily menu, it is possible to provide a reduced-salt diet without compromising palatability and calorie intake.

### 5.3. Association with Stress Conditions as a Quality of Life (QOL) Indicator During Salt-Reduction Implementation

As detailed above, appropriate salt intake plays an important role in maintaining homeostasis in human physiology, and overall adequate nutritional intake, including salt, is closely associated with health status. However, strict salt-reduction measures have been reported to lead to a decline in quality of life (QOL) and well-being, making long-term maintenance of salt-reduction efforts difficult [32,33]. In many cases, the extent of salt reduction and decreased QOL may be closely linked, but their causal relationship has not been fully investigated. This section provides a summary focusing on intervention studies that evaluate the intake of free-form L-Glu as an umami substance in meals, assess sodium intake during salt-reduction interventions via urinary sodium excretion, and examine stress markers in saliva (Table 1).

In the report by Iwamoto et al., a single-blind crossover intervention study was conducted with 31 healthy 18–35 year old adult female volunteers to assess three types of test diets: a normal diet, a reduced-salt diet (57% reduction), and a reduced-salt diet with added L-Glu as umami substance (56% reduction) [77]. As a result, a reduction in salt intake of more than 50%, as judged by the measurement of Na in urine, was achieved with the reduced-salt diet, regardless of the presence of added L-Glu. On the other hand, the stress marker chromogranin A (CgA) in saliva was significantly increased with the reduced-salt diet, although remaining at a similar level when compared to baseline value when the reduced-salt diet included added umami substances (see Figure 7A). Additionally, no changes were observed in energy intake or BMI due to the reduced-salt intervention. This suggests that the stress increase associated with strict salt reduction can be alleviated by the addition of Glu as an umami substance [77].

From a different perspective, the report by Hien et al. involved a 6-week crossover intervention trial with 42 adult volunteers (mean age 61.8 years, 45.2% female) [78]. Participants were divided into two groups: one group used seasonings as usual (Normal-Glu group) and the other group avoided seasonings containing high amounts of L-Glu as much as possible (Low-Glu group). The sodium-reduction guidance, which aimed at reducing the daily intake of sodium to less than 2000 mg, is based on three items: (1) Raising awareness about the impact of hypertension on health; (2) Underlying the importance of low-sodium diets for the control of hypertension; and (3) Providing a list of foods rich in sodium. As a result, the sodium-reduction rate after the intervention, as calculated from the urinary Na level, was 21.7% (L-Glu: 0.66 g/day) in the Low-Glu group and 46.3% (L-Glu: 1.40 g/day) in the Normal-Glu group (see Figure 7B). In this study, there was no significant difference in the stress marker (CgA) in saliva between the two groups. This study indicates that providing educational guidelines to reduce sodium intake while allowing the use of any seasoning with L-Glu in its free form enables subjects to maintain a sodium-reduced diet for 6 weeks without increasing stress indicators. Interestingly, the results obtained indicate that the more L-Glu is consumed, the greater the measured reduction in salt consumption [74].

Thus, these results indicate that umami substances such as L-Glu help to alleviate the stress indicator CgA during middle-term sodium reduction. These data are of major interest, since they suggest a possible causal link between the ingestion of L-Glu and the rate of salt reduction without an increase in the stress indicator. However, further research is needed to explore this relationship in more detail.

## 6. Conclusions and Prospects for Applied Aspects

Efforts to reduce salt intake and maintain an appropriate level of consumption are widely recognized as beneficial for human health. The continuous progress and development of salt-reduction methods have thus brought great hope in terms of public health. Then, any dietary strategy that can prove to be safe and efficient for maintaining the quantity of ingested salt in the range of adequate intake is worth being taken into consideration and implemented. In such context, this review summarizes to what extent umami substances, such as L-Glu, can contribute to the reduction in excessive salt consumption. Such a beneficial effect is based on the available clinical trials ranging from sensory evaluations of clear solutions (one cup) to the effects of daily diet menus with L-Glu being substituted to NaCl. Importantly, since it has been reported that maintaining a long-term reduction in salt intake without compromising quality of life (QOL) or well-being is difficult, this review focuses on research related to salt-reduction methods utilizing umami substances as one of the insights and strategies to address this issue. In fact, as presented above, some studies suggest that by properly utilizing umami substances such as L-Glu, it is possible to reduce salt content without compromising palatability, while also potentially alleviating stress indicators associated with salt reduction. Thus, as shown in Figure 8, the creation of a low-salt diet that maintains palatability, by skillfully utilizing seasonings such as umami substances in flavoring and cooking methods, along with its continuous consumption and by ensuring appropriate energy intake, can lead to the habituation of low-salt intake in a virtuous cycle beneficial for health. At the same time, by promoting the cycle between the different items, contributions can be made to solve the societal challenge of reducing salt intake.

On the other hand, this review has inherent limitations. Although numerous studies, reports, and guidelines support the protective effects of salt intake restriction, the exact mechanisms through which it influences sodium intake-related disease prevention and treatment remain unclear. Most research focuses primarily on epidemiological approaches, while large-scale, randomized clinical trials (such as sodium restriction intervention studies) that could provide clearer answers are still lacking. In particular, there is currently a lack of studies examining the effects of low-salt dishes using umami substances when consumed as part of daily intake or over a prolonged period. The main reason for this is the difficulty in accurately measuring both the amount of free amino acids (which serve as umami substances consumed with food) and the sodium excretion in urine, which is assessed through a 24 h urine collection to evaluate salt intake. This review references the results of studies that have been evaluated using scientific methods, taking this point into consideration. Further studies are needed to strengthen the evidence and expand knowledge in this field. Specifically, clinical intervention studies (such as long-term trials conducted over several months and those involving a broad range of participants in terms of age, gender, and health status) are necessary to strengthen the evidence base and deepen our understanding. Additionally, from a new perspective, more research is needed to explore the mechanisms behind reducing salt intake with minimal stress, in order to establish low-salt habits while maintaining quality of life (QOL).

Finally, it is plausible that L-Glu, by being recognized by different taste receptors than those involved in Na recognition, can modulate the perception of taste by the central nervous system, thus explaining that L-Glu in umami reduces salt intake without significantly altering food ingestion and quality of life. Such a proposition based on the results obtained from different clinical trials allows us to envisage a reduction in salt intake in the long-term perspective. Then, through the advancement of research in this field, it is hoped that the establishment of a low-salt intake habit through methods utilizing umami substances will contribute to healthier lives for individuals.

## Figures and Tables

**Figure 1 nutrients-17-01684-f001:**
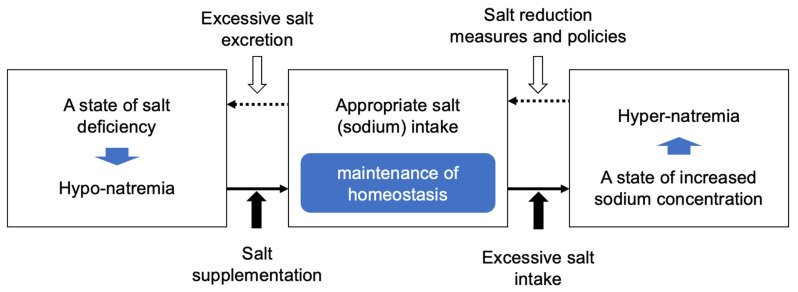
Diagram illustrating the need for appropriate salt intake.

**Figure 2 nutrients-17-01684-f002:**
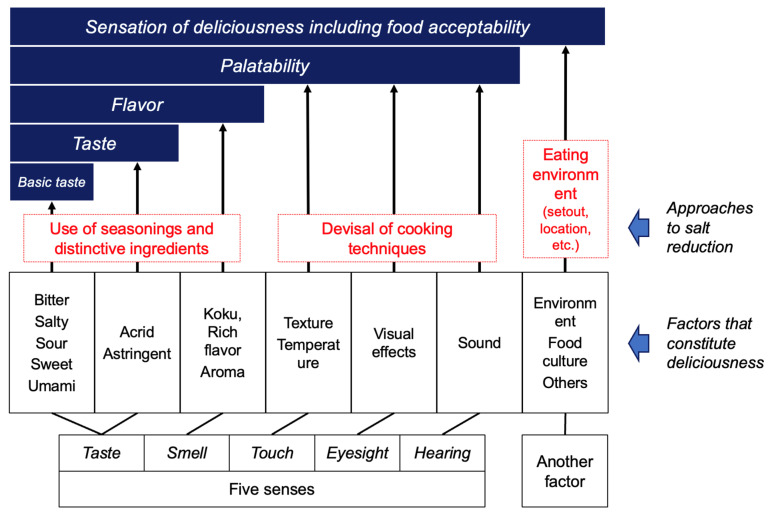
The structure of constitutive palatability factors and strategy for salt reduction. Created by the authors as inspired by the reference [40].

**Figure 3 nutrients-17-01684-f003:**
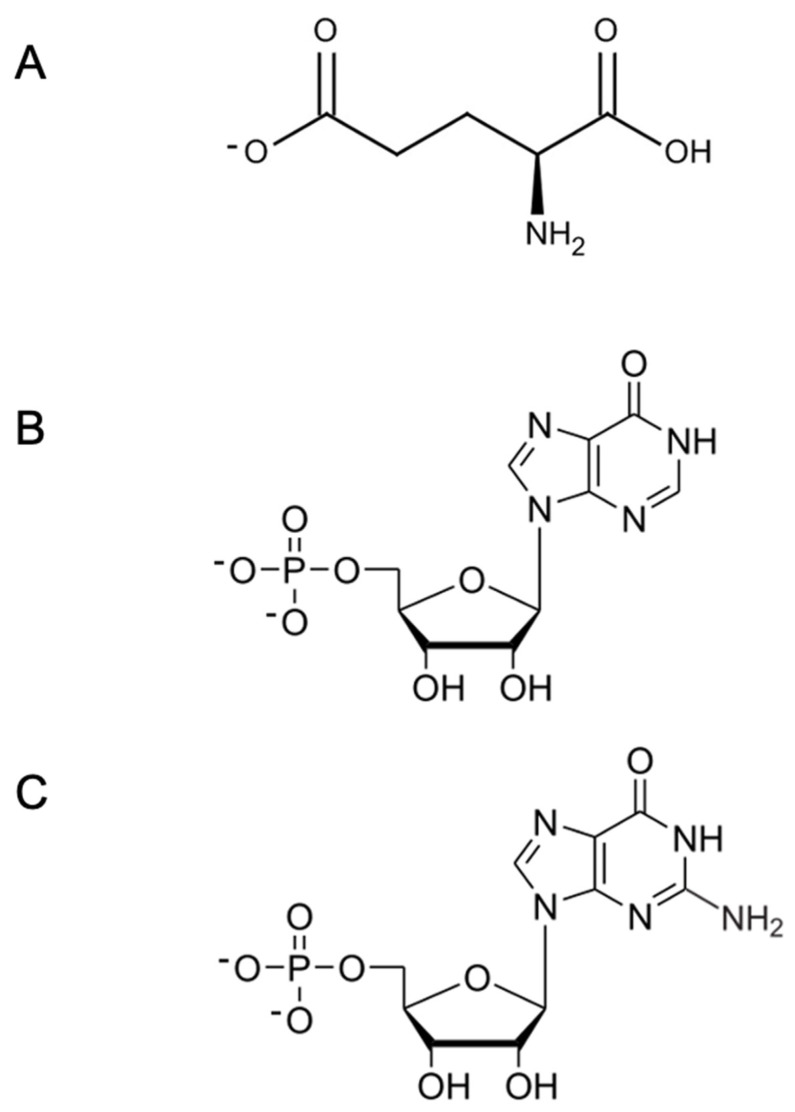
Chemical structure of main umami substances. (**A**) L-Glu, L-glutamate; (**B**) IMP, inosine 5′-monophosphate; and (**C**) GMP, guanosine 5′-monophosphate.

**Figure 4 nutrients-17-01684-f004:**
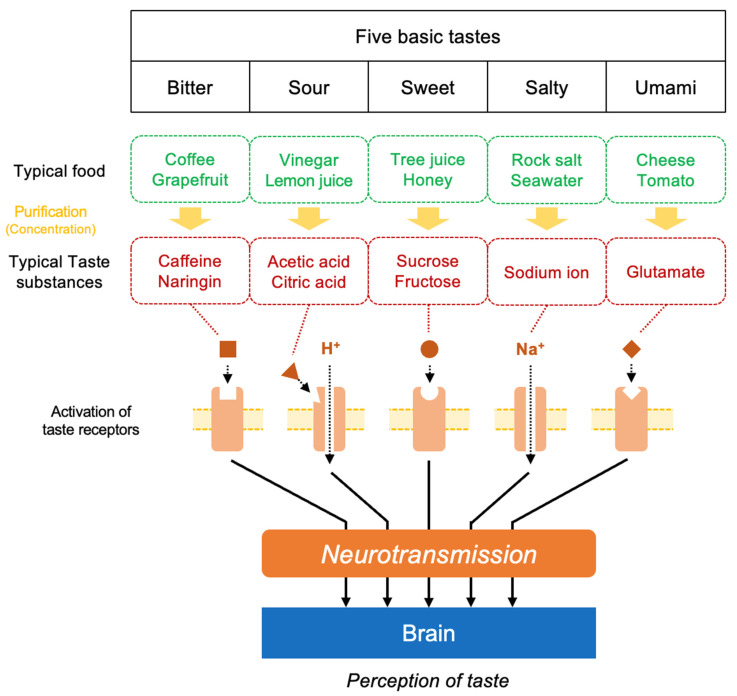
A schematic view of the structure of the signal transduction pathway and brain recognition for five basic tastes.

**Figure 5 nutrients-17-01684-f005:**
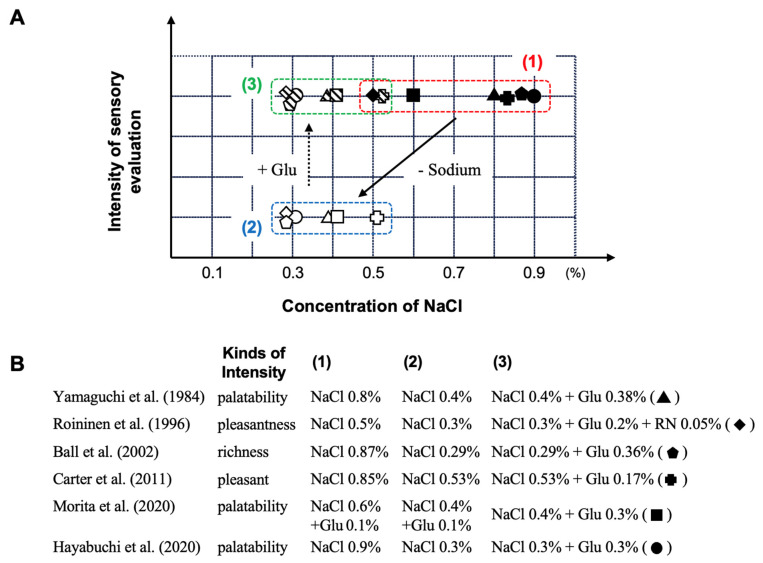
The influence of dietary sodium reduction on the palatability and capacity of Glu as an umami substance to restore this parameter. (1) original solution; (2) sodium-reduced solution; and (3) sodium-reduced solution + umami substance. (**A**): Sensory evaluation based on the measurement of intensity as a function of NaCl concentration with or without Glu; (**B**): published studies on that topic. Abbreviation: Glu, glutamate; NaCl, sodium chloride; RN, 5′-ribonucreotide as GMP and IMP. Created by the authors as inspired by the reference [48,70,71,72,73,74].

**Figure 6 nutrients-17-01684-f006:**
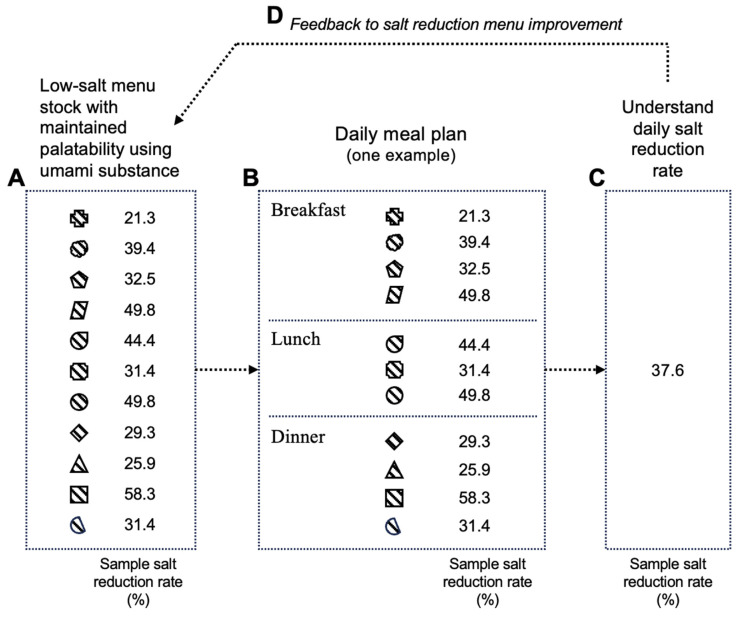
Image of salt-reduction menu for daily diet plan and feedback cycle of salt-reduction experience. A. Low-salt menu stock with maintained palatability using umami substance; B. Daily meal plan (one example); C. Understand daily salt reduction rate; D. Feedback to salt reduction menu improvement. Each symbol represents a meal menu with a different level of salt reduction. Created by the author as inspired by the reference [75,76].

**Figure 7 nutrients-17-01684-f007:**
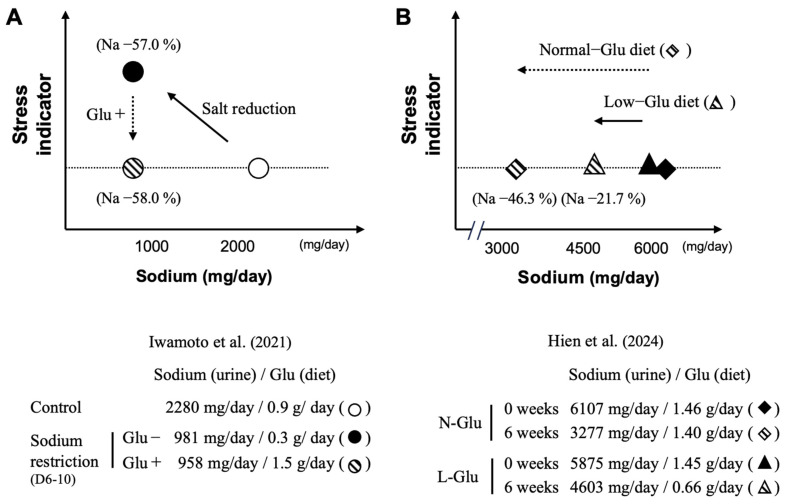
Stress indicator as a function of salt intake with and without L-glutamate. Strictly controlled salt restriction intervention study, with (added) or without (low) L-glutamate in diets (**A**); salt restriction intervention study, with (normal) or without (low) L-glutamate in a home cooking diet, following salt-reduction guidance (**B**). Normal-Glu, normal L-glutamate-using group; Low-Glu, low L-glutamate-using group. Created by the authors as inspired by the reference [77,78].

**Figure 8 nutrients-17-01684-f008:**
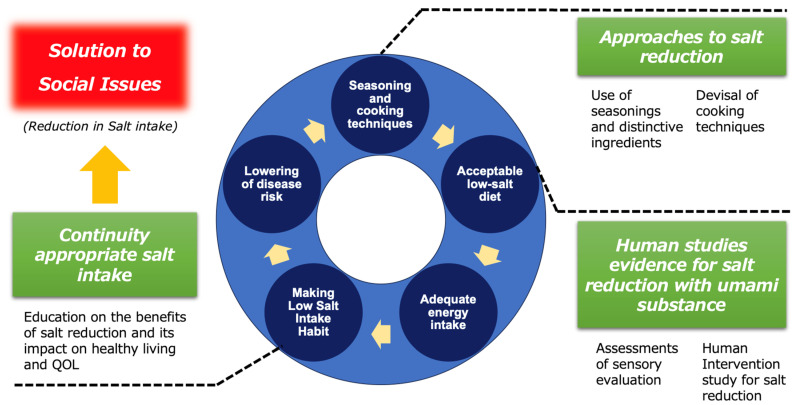
The conceptual model of a virtuous feedback cycle in long term salt-reduced diet intake.

**Table 1 nutrients-17-01684-t001:** Summary of characteristics of human studies for sodium reduction using umami substances.

Authors	Title	Study Design	Participants	Sample Size, Age	Outcome
Yamaguchi et al. [70]	Interactions of monosodium glutamate and sodium chloride on saltiness and palatability of a clear soup.	Sensory evaluation, soup	Healthy adults	*N* = 40 (females and males), 20–40 years	Sodium reduction, palatability
Roininen et al. [71]	Effect of umami taste on pleasantness of low-salt soups during repeated testing.	Sensory evaluation, soup	Volunteers (students and university staff members)	*N* = 44 (35 females, 9 males), 27 years	Sodium reduction, pleasantness
Ball et al. [72]	Calcium diglutamate improves taste characteristics of lower-salt soup.	Sensory evaluation, soup	Young adults (almost university students)	*N* = 120 (67% females, 33% males), 22 years	Sodium reduction, richness
Carter et al. [73]	The sensory optimum of chicken broths supplemented with calcium di-glutamate: a possibility for reducing sodium while maintaining taste.	Sensory evaluation, soup	Normal weight adults	*N* = 34 (females and males), 20–35 years	Sodium reduction, pleasant
Morita et al. [48]	Quantitative verification of the effect of using an umami substance (L-glutamate) to reduce salt intake.	Sensory evaluation, solution	Healthy adults	*N* = 11 (females), 30.5 ± 3.7 years	Sodium reduction, palatability
Hayabuchi et al. [74]	Validation of preferred salt concentration in soup based on a randomized blinded experiment in multiple regions in Japan influence of umami (L-glutamate) on saltiness and palatability of low-salt solutions.	Sensory evaluation, solution (multicenter trial)	Healthy adults	*N* = 584 (515 females, 69 males), Age: 19–29, *n* = 257 30–59, *n* = 72 60–69, *n* = 100 >70, *n* = 153	Sodium reduction, palatability
Ishida et al. [75]	Sensory evaluation of a low-salt menu created with umami, similar to savory, substance.	Sensory evaluation, hospital meals	Healthy adults	*N* = 34 (20 females, 14 males), 41.9 ± 15.2 years	Sodium reduction, palatability
Kawano et al. [76]	Pilot intervention study of a low-salt diet with Monomagnesium di-L-glutamate as an umami seasoning in psychiatric inpatients.	Intervention study	Psychiatric inpatients	*N* = 15 (11 females, 4 males), 52.9 ± 15.3 years	Sodium reduction, energy intake
Iwamoto et al. [77]	Stress condition on a restricted sodium diet using umami substance (L-Glutamate) in a pilot randomized cross-over study.	Intervention study	Volunteers (students and university staff members)	*N* = 31 (females), 18–35 years	Sodium reduction, stress marker
Hien et al. [78]	Dietary free L-glutamate contributes to maintaining a low sodium intake among Vietnamese.	Intervention study	Prehypertension and urinary Na ≥ 4000 mg/day	*N* = 42 (females), 61.8 ± 15.3 * years	Sodium reduction, stress marker

Note: Ages are indicated as mean ± SD.; * are indicated as SE. Abbreviation: Na, sodium; SD, standard deviation; SE, standard error.

## Data Availability

The data presented in this study are available in the article.

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
