# Peer review of "The Role of L-Glutamate as an Umami Substance for the Reduction of Salt Consumption: Lessons from Clinical Trials"

_nutrients, 2025, doi:10.3390/nu17101684_

Round 1
Reviewer 1 Report
Comments and Suggestions for Authors
The article addresses a specific and timely gap in nutritional science and public health related to strategies for salt reduction while maintaining palatability and compliance. The paper focuses on the challenge of sustaining reduced salt intake in diets, which is crucial for preventing non-communicable diseases like hypertension. Based on the article here are specific methodological improvements and further controls that the authors should consider:
- Most trials referenced were short-term. Future studies should implement longitudinal designs to better assess the sustainability of reduced salt intake with glutamate over months.
- Several studies focus on narrow participant groups. Future research should ensure greater demographic diversity in age, gender and health status.
- Baseline dietary sodium intake was not always well-controlled or reported. A run-in period with standardized sodium intake across all participants would improve the internal validity of the findings.
Implementing these improvements and controls would enhance the study's rigor, validity, and policy implications.
Reviewer 2 Report
Comments and Suggestions for Authors
The study by Hideki Matsumoto et al. summarized recent advances in the study of the role of glutamate as umami substance for the reduction of salt consumption. The review is informative and interesting. I have the following questions and comments.
1, the resolution of figure 3 to figure 8 is very low. Please revise. All these figures need to be replaced.
2, for a review study, the authors must specify how the literature was searched, the keywords used for the collection of the studies. The reason why some studies were included and why others were excluded should also be discussed.
3, are there any side effects if we replace salt with glutamate in our daily diet? This must be discussed.
4, some of the paragraphs in the manuscript is too long. The authors may consider divide them into different parts. For example, line 35 to 56, line 224 to 246.
5, I suggest the authors to add a new table to summarize the most important findings in different human clinical trails. This would help the readers to understand the study more easily.
Reviewer 3 Report
Comments and Suggestions for Authors
The manuscript presents an interesting narrative review devoted to the use of glutamate as an umami substance to reduce the sodium uptake. This question is actual and important. The review is generally well and competently written. The conclusions and practical prospects are scientifically sound.
Lines 5-8: Dear Authors, please write the names of the Department and Faculty starting words with capital letters, like in lines 9-10
Line 5: Start from the lower unit: first department, then faculty
Lines 39-41: If to be precise, is NaCl or Na+ involved in the processes listed?
Figure 1: Please correct “Hypermatremia” to “Hypernatremia”
Line 92: “the capacity of Glu”, I propose “the capacity of glutamate (Glu)”
Lines 107/108:” Created by the author“, there are multiple authors so it would be more appropriate to write “by the authors” or add initials after “the author”; the same refers to Figures 5 and 7
Lines 149/150:” Glu receptors in the oral cavity (taste bud) and the mucosa of the digestive tract”, I would suggest “Glu receptors in the oral cavity (in taste buds) and in the mucosa of the digestive tract”
Figure 4: “Five basic taste”, please correct to “Five basic tastes”
Line 209:” 5’-ribonucreotide”, which 5’-ribonucleotide?
Line 271: “CgA”, please explain the acronym on the first use
Abbreviations: Please write institution names: each word in a capital letter
Please format References according to the requirements of the journal
Comments on the Quality of English LanguageAll words in the names of the Department and Faculty (Affiliations) should start words with capital letters.The same referrs to the names of institutions (Abbreviations)
Round 2
Reviewer 2 Report
Comments and Suggestions for Authors
The authors the revised the manuscript accordingly. It can be considered for publication.